# Differences in Geochemical Characteristics and Tectonic Settings between Hai Van Granitic Rocks in Da Nang Province and Van Canh Plutonic Rocks in Quang Nam Province, Central Vietnam

**Etsuo Uchida *** and **Takumi Yokokura**

Department Resource and Environmental Engineering, Faculty of Science and Engineering, Waseda University, Ohkubo 3-4-1, Shinjuku, Tokyo 169-8555, Japan; tyokokura@toki.waseda.jp
* Correspondence: weuchida@waseda.jp

**Abstract:** Research was conducted on plutonic rocks, previously referred to as the Hai Van granitic rocks, distributed in Da Nang and Quang Nam provinces in Central Vietnam. The granitic rocks in Da Nang Province have low magnetic susceptibilities and have geochemical signatures typical of S-type. Additionally, a negative Eu anomaly suggests that the source rock is an organic matter-bearing sedimentary rock. The granitic rocks were likely formed during the collision between the Indochina and South China blocks. In contrast, plutonic rocks in Quang Nam Province have high magnetic susceptibilities and have geochemical signatures of I-type. No Eu anomaly was observed, and they are adakitic rocks in nature. Based on these findings, the plutonic rocks in Quang Nam Province are distinctly different from the Hai Van granitic rocks in Da Nang Province, but they are Van Canh plutonic rocks. The Hai Van granitic rocks in Da Nang Province and the Van Canh plutonic rocks in Quang Nam Province are located in the Truong Son Fold Belt. The Van Canh plutonic rocks are located farther away from the Song Ma Suture Zone than the Hai Van granitic rocks. The Van Canh plutonic rocks were generated due to the subduction of the hot Song Ma Ocean beneath the Indochina Block. The Hai Van granitic rocks are understood to have been generated in a compressional field where the Song Ma Ocean was pushing against the Indochina Block; however, the Van Canh plutonic rocks are supposed to have been generated in an extensional field, like in a back-arc-like environment generated by the subduction of the Song Ma Ocean beneath the Indochina Block.

**Keywords:** Hai Van granitic rock; Van Canh plutonic rock; Truong Son Fold Belt; Song Ma Ocean; Indochina Block; South China Block; geochemical signature; adakitic rock; Vietnam

## 1. Introduction

In Da Nang and Quang Nam provinces, located in the central part of Vietnam, a significant distribution of plutonic rocks can be observed (Figure 1). Hieu et al. [1] conducted a study on Hai Van granitic rocks in this region. In the current study, we conducted a sampling of plutonic rocks in a similar region to that investigated by Hieu et al. [1]. We conducted in situ magnetic susceptibility measurements and carried out the chemical analysis of whole-rock composition on the plutonic rocks. Additionally, we carried out a chemical composition analysis of the biotite in the plutonic rocks and measured Nd and Sr isotope ratios. The results demonstrate that in addition to the Hai Van granitic rocks, Van Canh plutonic rocks are also mixed with the granitic rocks studied by Hieu et al. [1]. Furthermore, based on the results of this investigation, we discuss the formation mechanisms of both plutonic rock suites in the surveyed region. These two rock bodies are located within the Truong Son Fold Belt. Another objective of this study is to elucidate the formation mechanism of the Truong Son Fold Belt based on the difference in the formation process of the two rock bodies.

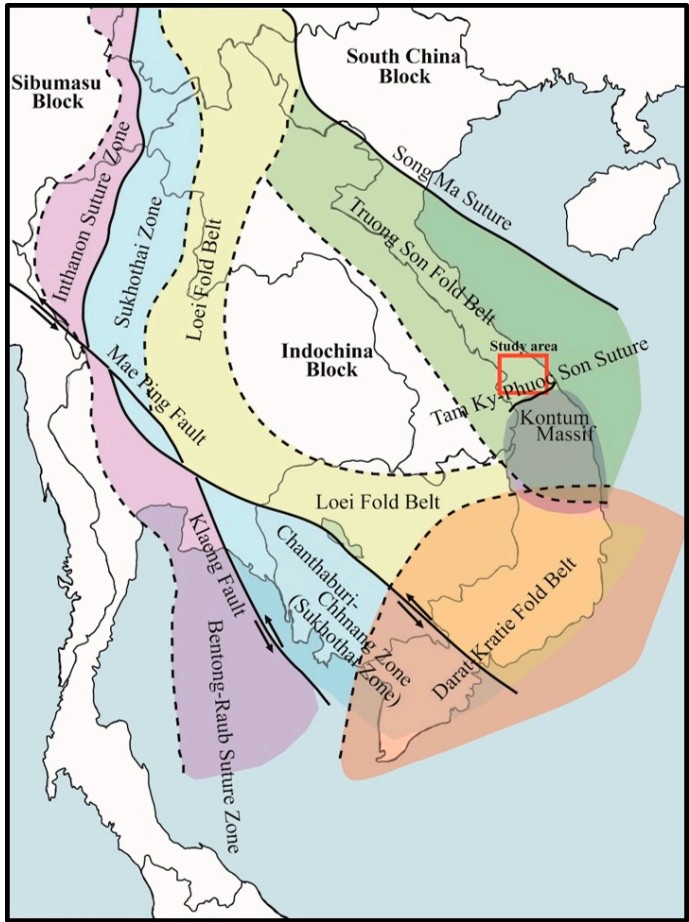

**Figure 1.** Map showing tectonic settings in Southeast Asia [2–7]. A red square shows the study area. The arrows indicate the direction of displacement along the faults.

## 2. Geological Settings

Vietnam has an elongated shape, stretching approximately 1600 km in a north–south direction. It is divided by the Song Ma Suture Zone. The north side belongs to the South China Block, and the south side to the Indochina Block (Figure 1) [8–15]. The south side is divided into three areas from north to south: the Truong Son Fold Belt, Kontum Massif, and Dalat-Kratie Fold Belt, although their boundaries are not well defined. The Truong Son Fold Belt is believed to extend to the southern part of the Kontum Massif [16]. Additionally, Cretaceous granitic rocks distributed in the Dalat-Kratie Fold Belt are thought to extend as far as the Kontum Massif. Therefore, the boundaries between the Truong Son Fold Belt, Kontum Massif, and Dalat-Kratie Fold Belt overlap with each other.

The study area belongs to the southern part of the Truong Son Fold Belt (Figure 1). The Truong Son Fold Belt was formed by the subduction of the Song Ma Ocean, a branch of the Paleo-Tethys Ocean, between the South China and Indochina Blocks. This area is believed to have experienced two episodes of continental arc magmatism during the Early Permian (ca. 290–260 Ma) and Triassic (ca. 245–230 Ma) [6].

We conducted sampling of the Hai Van granitic rocks at Hai Van Pass and Cape Da Nang in Da Nang Province. We also collected plutonic rocks in the Ben Giang and Que Son districts in Quang Nam Province, which appear to be the Van Canh plutonic rocks rather than the Hai Van granitic rocks (Figure 2) [17].

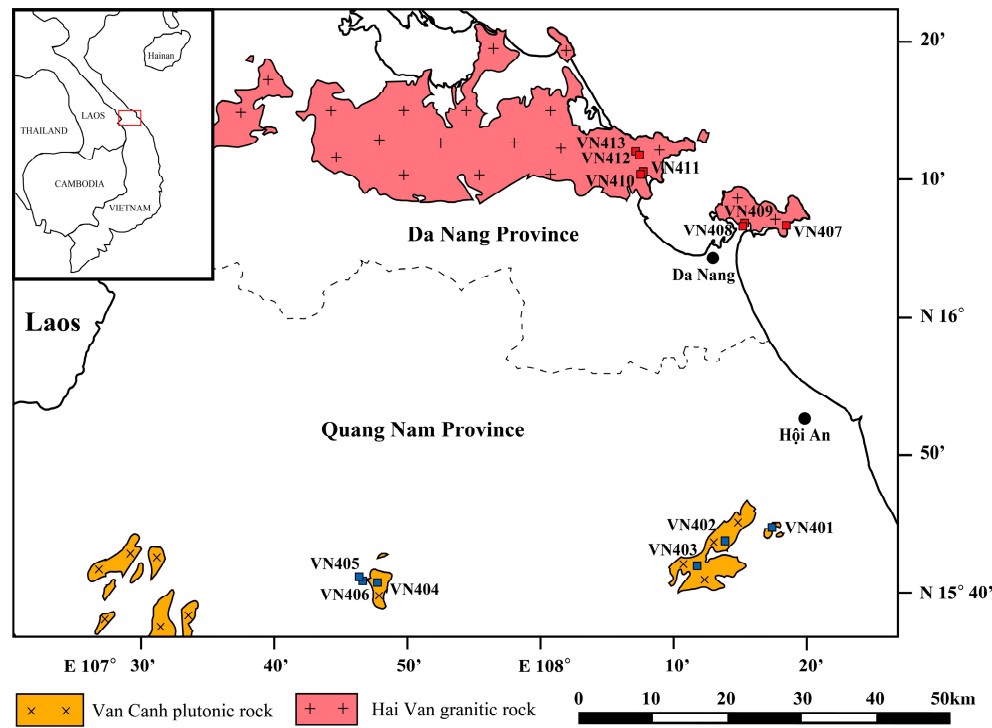

**Figure 2.** Distribution map of the Hai Van granitic rocks in Da Nang Province and the Van Canh plutonic rocks in Quang Nam Province, showing the sampling locations of the Hai Van granitic rocks (red square) and the Van Canh plutonic rocks (blue square). The map is drawn on the basis of the Geological and Mineral Resources Map of Vietnam [17].

## 3. Materials and Methods

We conducted plutonic rock sampling in a similar region to that investigated by Hieu et al. [1] (Table 1 and Figure 2). The collection of plutonic rock samples was carried out at roadside outcrops or quarries by visually selecting unaltered rocks. We collected three plutonic rock samples in Cape Da Nang, four in Hai Van Pass in Da Nang Province, and three each in the Ben Giang and Que Son districts in Quang Nam Province (Figure 2). At the sampling sites, magnetic susceptibility measurement was performed using a magnetic susceptibility meter (SM30, ZH Instruments, Brno, Czech Republic) in a basic mode. Measurements were taken at 10 locations, choosing unaltered and flat rock surfaces.

**Table 1.** Latitudes and longitudes of sampling locations of the plutonic rocks.

| Granitic Body | Sample No. | Latitude | Longitude |
|---|---|---|---|
| | VN401 | 15°44′42.3″ N | 108°17′12.7″ E |
| | VN402 | 15°43′51.5″ N | 108°14′17.1″ E |
| Van Canh | VN403 | 15°42′01.3″ N | 108°11′17.1″ E |
| plutonic rock | VN404 | 15°40′34.4″ N | 107°47′47.4″ E |
| | VN405 | 15°41′02.0″ N | 107°46′24.4″ E |
| | VN406 | 15°40′38.1″ N | 107°46′48.0″ E |
| | VN407 | 16°06′26.1″ N | 108°18′29.0″ E |
| | VN408 | 16°06′25.1″ N | 108°15′23.2″ E |
| Hai Van | VN409 | 16°06′42.8″ N | 108°15′27.7″ E |
| granitic rock | VN410 | 16°10′09.4″ N | 108°07′34.1″ E |
| | VN411 | 16°10′21.1″ N | 108°07′47.2″ E |
| | VN412 | 16°11′30.9″ N | 108°07′29.8″ E |
| | VN413 | 16°12′01.6″ N | 108°06′46.6″ E |

For all collected plutonic rock samples, thin sections were prepared, and constituent mineral identification was conducted using transmitted light polarizing microscopy.

The samples were pulverized for 1 min by a vibrating mill made of tungsten carbide (TI-100, Heiko Seisakusho Ltd., Fukushima, Japan). The pulverized samples were further ground for approximately 10 min using an agate mortar. Approximately 5 g of these prepared rock powders were sent to Activation Laboratories Ltd. (Ancaster, ON, Canada). An analysis of whole-rock chemical composition was requested based on the 4Litho package. The analysis included 55 elements, excluding ignition loss. The rock powder was fused with lithium metaborate/tetraborate and dissolved with a dilute nitric acid solution. The analysis was performed by combining inductively coupled plasma–optical emission spectrometry (ICP-OES) and inductively coupled plasma–mass spectrometry (ICP-MS). However, due to the use of a tungsten carbide mill during the rock pulverization, there was contamination with W and Co. As a result, these elements were excluded from the analysis results.

The analysis of biotite was carried out using scanning electron microscopy (SEM; JEOL JSM-6360, Tokyo, Japan), which was equipped with an energy-dispersive X-ray spectrometer (EDS; INCA-ENERGY, Oxford Instruments, Abingdon, UK). Prior to the measurement, observation was carried out using a polarizing microscope, and unaltered biotite was selected for analysis. Carbon coating was performed using a carbon coater (SC-701C, Sanyu Denshi, Tokyo, Japan) before the SEM-EDS analysis. In the SEM-EDS analysis, the acceleration voltage was set at 15 keV. The current was adjusted to achieve a total count rate of 2000 counts/s on the metal cobalt surface. The measured elements were Si, Ti, Al, Fe, Mg, Mn, K, and Na, with the oxygen content determined stoichiometrically. A valence of 2 was assumed for Fe and Mn. Synthetic $SiO_2$, $TiO_2$, $Al_2O_3$, $Fe_2O_3$, and MnO, as well as natural potassium feldspar and albite, were used as standards.

Nd and Sr isotope ratios were determined on collected plutonic rocks at the Research Institute for Humanity and Nature. Approximately 100 mg of powdered sample was decomposed at 160 °C in a container made of polytetrafluoroethylene, along with nitric acid ($HNO_3$; 0.7 mL), perchloric acid ($HClO_4$; 0.2 mL), and hydrofluoric acid (HF; 1.0 mL). The extraction of Nd and Sr was achieved using Ln and Sr resins, respectively (Eichrom Technologies Inc., University Lane Lisle, IL, USA). The isotope ratio measurements of separated Nd and Sr were performed using a multi-collector inductively coupled plasma–mass spectrometer (NEPTUNE, Thermo Fisher Scientific Inc., Waltham, MA, USA). Measured $^{143}Nd/^{144}Nd$ and $^{87}Sr/^{86}Sr$ ratios were normalized with respect to $^{146}Nd/^{144}Nd = 0.7219$ and $^{86}Sr/^{88}Sr = 0.1194$ (i.e., the ratios existing in nature), respectively [18]. Repeated analyses of the standards JNdi-1 and NIST SRM 987 yielded an average $^{143}Nd/^{144}Nd$ ratio of $0.512048 \pm 0.000023$ (2σ) (n = 5) and $^{87}Sr/^{86}Sr$ ratio of $0.710299 \pm 0.000033$ (2σ) (n = 4), respectively. The isotope ratios of Nd and Sr in the samples were corrected with respect to standard samples: $^{143}Nd/^{144}Nd = 0.512115$ for JNdi-1 [18] and $^{87}Sr/^{86}Sr = 0.710250$ for NIST SRM 987 [19], respectively.

## 4. Results

### 4.1. Description of the Collected Plutonic Rocks

Figure 3 shows photographs of representative samples collected in Da Nang and Quang Nam provinces, and photomicrographs of their thin sections under a polarizing microscope. Table 2 summarizes the constituent minerals of the collected plutonic rocks.

The constituent particles of the Van Canh plutonic rocks in Quang Nam Province are coarse-grained (Figure 3), consisting primarily of minerals such as quartz, plagioclase, potassium feldspar, biotite, and hornblende, along with accessory minerals including apatite, titanite, zircon, opaque minerals, and epidote (Table 2).

In contrast, the granitic rocks in Da Nang Province comprise medium- to fine-grained constituent particles (Figure 3), and primarily consist of minerals such as quartz, plagioclase, potassium feldspar, biotite, and muscovite, with accessory minerals such as apatite, zircon, and opaque minerals (Table 2).

# Van Canh plutonic rock

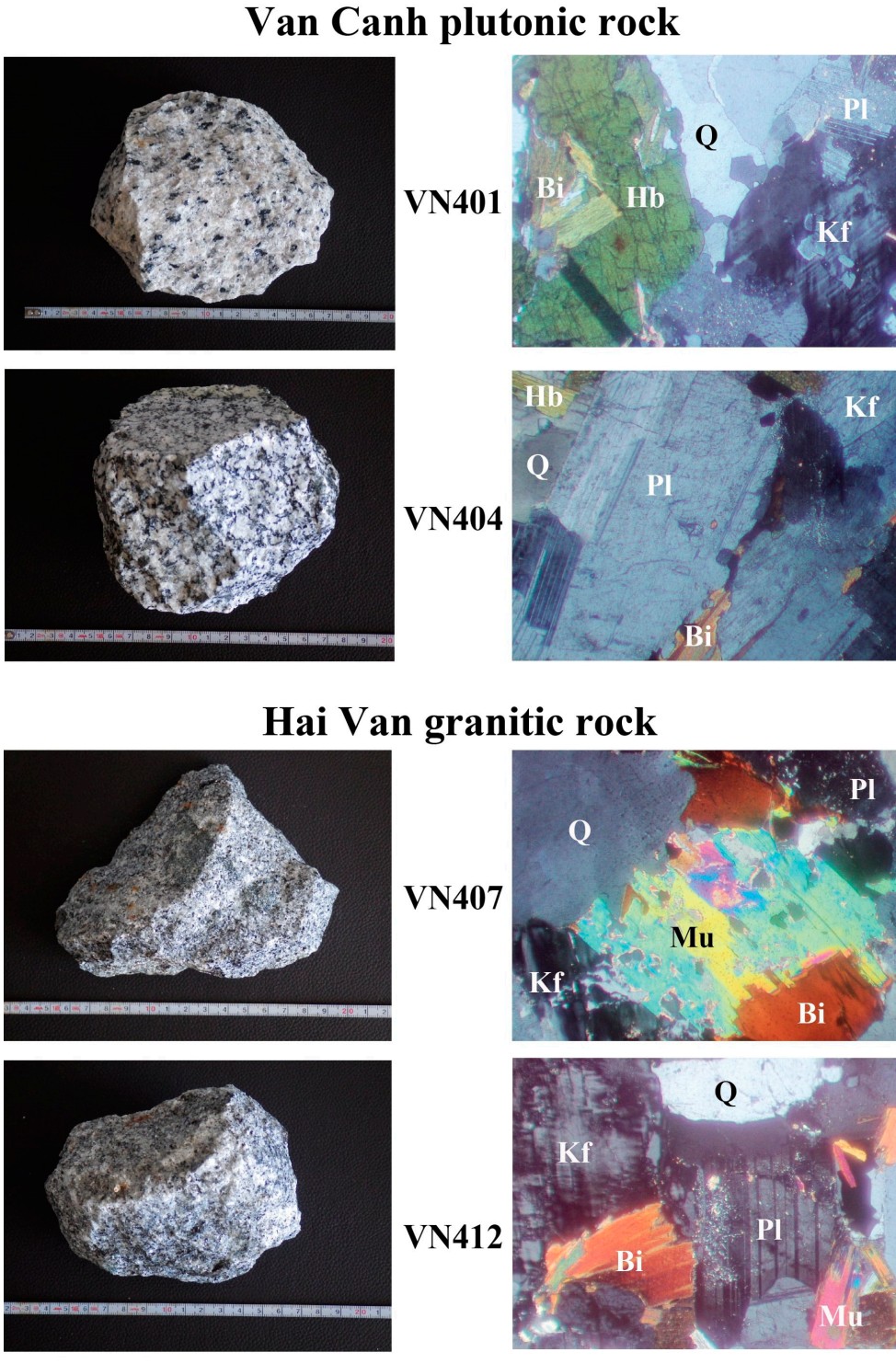

# Hai Van granitic rock

**Figure 3.** Representative rock samples (**left**) from the Van Canh plutonic rocks and the Hai Van granitic rocks, and their photomicrographs (**right**) under a transmission cross-polarized microscope. Abbreviations: Pl, plagioclase; Kf, potassium feldspar; Q, quartz, Bi, biotite; Hb, hornblende; Mus, muscovite.

**Table 2.** Rock classification and constituent minerals for the collected samples.

| Granitic Body | Sample No. | Rock Type | Q | Pl | Kf | Bi | Hb | Zr | Ap | Mu | Ti | Op | Ep | Ru | Cpx | Tour | Alla | Cal | Remarks |
|---|---|---|---|---|---|---|---|---|---|---|---|---|---|---|---|---|---|---|---|
| Van Canh plutonic rock | VN401 | Hornblende-Biotite granite | ◎ | ○ | ◎ | △ | △ | – | – | – | | | – | | | | | | Biotite is altered. |
| | VN402 | Hornblende-Biotite granite | ◎ | ○ | ◎ | – | △ | – | – | – | – | – | | | | | | | Biotite is altered. |
| | VN403 | Synite-Diorite | ◎ | ○ | ○ | △ | △ | – | – | | – | – | | | | | | | |
| | VN404 | Diorite | ○ | ◎ | ○ | △ | △ | – | – | | – | – | | | | | | | |
| | VN405 | Synite-Diorite | | ◎ | | | △ | – | – | | – | – | – | | | | | | Plagioclase is altered. |
| | VN406 | Diorite | ○ | ◎ | △ | | △ | – | – | | – | – | | | | | | | |
| Hai Van granitic rock | VN407 | Biotite granite | ◎ | ◎ | △ | ○ | | – | – | △ | | – | | | | | | | |
| | VN408 | Biotite granite | ◎ | ○ | – | ○ | | – | – | – | | – | | | | | | | |
| | VN409 | Biotite granite | ○ | ○ | △ | ○ | | – | – | △ | | – | | | | | | | |
| | VN410 | Biotite granite | ○ | ○ | △ | ○ | | – | – | △ | | – | | | | | | | |
| | VN411 | Biotite granite | ○ | ○ | ○ | △ | | – | – | △ | | – | | | | | | | |
| | VN412 | Biotite granite | ○ | ○ | ○ | △ | | – | – | ○ | | – | | – | | | | | |
| | VN413 | Biotite granite | ○ | ○ | ○ | △ | | – | – | △ | | – | | | | | | | K-feldspar is altered. |

Modal proportions: ◎, 50–30 vol.%; ○, 30–10 vol.%; △, 10–2 vol.%; –, less than 2 vol.%. Abbreviations: Kf, potassium feldspar; Pl, plagioclase; Q, quartz; Bi, biotite; Hb, hornblende; Mu, muscovite; Cpx, clinopyroxene; Ap, apatite; Zr, zircon; Ti, titanite; Ep, epidote; Cpx, clinopyroxene; Ru, rutile; Alla, allanite; Tour, tourmaline; Cal, calcite; Op, opaque minerals.

### 4.2. Magnetic Susceptibility

Ishihara [20] classified granitic rocks into two groups: magnetite series and ilmenite series. According to Ishihara [21], granitic rocks are classified based on magnetic susceptibility, and rocks with magnetic susceptibility larger than $3 \times 10^{-3}$ SI are classified as magnetite series.

The Van Canh plutonic rocks investigated in Quang Nam Province, except for one sample, exhibit magnetic susceptibilities larger than $3 \times 10^{-3}$ SI and can be classified as magnetite series (Figure 4) [20,21]. In contrast, the Hai Van granitic rocks in Da Nang Province all had magnetic susceptibilities lower than $3 \times 10^{-3}$ SI, indicating their affiliation with the ilmenite series.

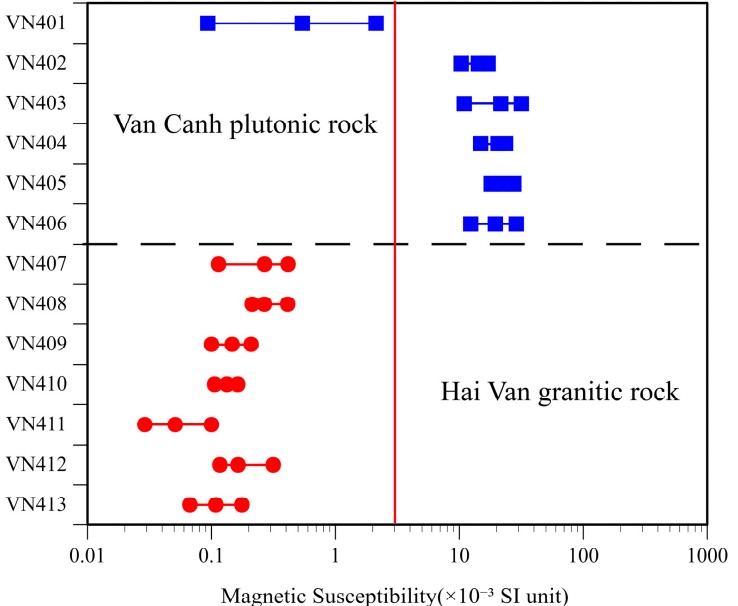

**Figure 4.** Results of magnetic susceptibility measurements (minimum, mean, and maximum values). Blue squares, the Van Canh plutonic rocks; red circles, the Hai Van granitic rocks.

### 4.3. Whole-Rock Chemical Composition

The results of the whole-rock chemical composition analysis for the investigated plutonic rocks are presented in Table 3.

The $Na_2O + K_2O$ vs. $SiO_2$ (TAS) diagram is shown in Figure 5 [22,23]. From this diagram, it is evident that the Hai Van granitic rocks in Da Nang Province are predominantly classified as granite. In contrast, the Van Canh plutonic rocks in Quang Nam Province have a wide range of $SiO_2$ contents and can be categorized as granite, diorite, and syenite-diorite. Among the plutonic rocks from both regions, three rocks can be classified as alkalic rocks.

Based on the $Na_2O$ vs. $K_2O$ diagram (Figure 6), the Hai Van granites in Da Nang Province exhibit relatively low $Na_2O$ content, and they are primarily classified as S-type [24]. In contrast, the Van Canh plutonic rocks in Quang Nam Province have relatively high $Na_2O$ content, classifying them as I-type.

The Hai Van granitic rocks in Da Nang Province are richer in incompatible elements such as Nb, Ta, Th, Pb, and U than the Van Canh plutonic rocks in Quang Nam Province.

The $Al_2O_3/(Na_2O + K_2O)$ vs. $Al_2O_3/(CaO + Na_2O + K_2O)$ diagram (A/NK vs. A/CNK diagram) (Figure 7) reveals that the Hai Van granitic rocks in Da Nang Province have A/CNK ratios higher than 1.1, indicating peraluminous characteristics and aligning with S-type granitic rocks, which is consistent with the $Na_2O$ vs. $K_2O$ diagram results [24,25]. In contrast, the Van Canh plutonic rocks in Quang Nam Province have A/CNK ratios lower than 1.1, falling within the metaluminous to peraluminous range, classifying them as I-type.

Table 3. Results of the whole-rock chemical composition analysis.

| Sample No. | | Van Canh Plutonic Rock | | | | | | Hai Van Granitic Rock | | | | | | |
|---|---|---|---|---|---|---|---|---|---|---|---|---|---|---|
| | | VN401 | VN402 | VN403 | VN404 | VN405 | VN406 | VN407 | VN408 | VN409 | VN410 | VN411 | VN412 | VN413 |
| $SiO_2$ | % | 74.05 | 74.46 | 62.96 | 62.12 | 51.38 | 61.51 | 71.59 | 69.04 | 70.19 | 70.9 | 73.25 | 70.02 | 71.18 |
| $Al_2O_3$ | % | 12.14 | 12.86 | 18.07 | 18.24 | 19.12 | 18.67 | 13.49 | 13.91 | 13.81 | 13.43 | 13.24 | 13.56 | 12.93 |
| $Fe_2O_3(T)$ | % | 1.99 | 1.57 | 3.47 | 4.91 | 8.33 | 4.53 | 4.34 | 5.4 | 4.47 | 3.61 | 1.7 | 4.42 | 4.33 |
| MnO | % | 0.026 | 0.054 | 0.052 | 0.091 | 0.166 | 0.097 | 0.083 | 0.091 | 0.057 | 0.077 | 0.036 | 0.06 | 0.063 |
| MgO | % | 0.12 | 0.24 | 0.84 | 2.01 | 3.09 | 1.75 | 1.53 | 2.09 | 1.7 | 1.08 | 0.23 | 1.56 | 1.71 |
| CaO | % | 1.23 | 1.62 | 3.33 | 6.37 | 7.77 | 5.22 | 2.31 | 2.09 | 1.13 | 1.96 | 0.87 | 1.22 | 1.2 |
| $Na_2O$ | % | 3.12 | 3.23 | 5.03 | 4.05 | 3.77 | 4.53 | 2.66 | 2.59 | 1.76 | 2.34 | 2.5 | 1.67 | 1.38 |
| $K_2O$ | % | 4.8 | 4.02 | 3.28 | 0.79 | 1.64 | 1.62 | 2.65 | 2.73 | 3.9 | 3.53 | 5.62 | 3.68 | 3.46 |
| $TiO_2$ | % | 0.139 | 0.119 | 0.41 | 0.658 | 0.941 | 0.569 | 0.544 | 0.708 | 0.545 | 0.501 | 0.204 | 0.531 | 0.483 |
| $P_2O_5$ | % | 0.02 | 0.02 | 0.15 | 0.17 | 0.34 | 0.17 | 0.37 | 0.23 | 0.08 | 0.1 | 0.06 | 0.16 | 0.16 |
| LOI | % | 1 | 0.75 | 0.69 | 1.03 | 2.09 | 1.61 | 1.07 | 1.47 | 1.93 | 0.9 | 0.81 | 1.5 | 1.86 |
| Total | % | 98.63 | 98.93 | 98.28 | 100.4 | 98.65 | 100.3 | 100.6 | 100.3 | 99.58 | 98.42 | 98.52 | 98.39 | 98.75 |
| Sc | ppm | 7 | 3 | 4 | 6 | 10 | 6 | 12 | 15 | 12 | 10 | 5 | 10 | 10 |
| Be | ppm | 2 | 2 | 5 | 2 | 1 | 2 | 2 | 2 | 3 | 1 | 6 | 4 | 5 |
| V | ppm | 6 | 9 | 37 | 84 | 132 | 69 | 74 | 91 | 74 | 56 | 9 | 65 | 57 |
| Ba | ppm | 241 | 1022 | 894 | 413 | 601 | 634 | 670 | 316 | 775 | 647 | 310 | 628 | 413 |
| Sr | ppm | 83 | 141 | 486 | 658 | 746 | 755 | 147 | 140 | 109 | 104 | 53 | 120 | 50 |
| Y | ppm | 41 | 16 | 25 | 8 | 17 | 9 | 39 | 30 | 19 | 40 | 47 | 25 | 45 |
| Zr | ppm | 215 | 87 | 284 | 171 | 171 | 152 | 146 | 183 | 189 | 220 | 153 | 198 | 170 |
| Cr | ppm | <20 | <20 | <20 | <20 | <20 | <20 | 60 | 70 | 60 | 30 | <20 | 60 | 50 |
| Ni | ppm | <20 | <20 | <20 | <20 | <20 | <20 | 30 | 40 | <20 | <20 | <20 | 30 | 20 |
| Cu | ppm | <10 | <10 | <10 | <10 | 30 | <10 | 40 | 20 | 10 | 20 | <10 | 20 | <10 |
| Zn | ppm | 50 | 30 | 90 | 60 | 110 | 60 | 70 | 90 | 70 | 60 | 30 | 80 | 60 |
| Ga | ppm | 18 | 14 | 28 | 21 | 25 | 22 | 17 | 19 | 19 | 17 | 19 | 20 | 18 |
| Ge | ppm | < 1 | 1 | 1 | 1 | 2 | 1 | 1 | 2 | 2 | 2 | 2 | 2 | 1 |
| As | ppm | <5 | <5 | <5 | <5 | 6 | <5 | <5 | <5 | <5 | <5 | <5 | <5 | <5 |
| Rb | ppm | 102 | 167 | 142 | 22 | 57 | 45 | 130 | 166 | 229 | 197 | 437 | 225 | 248 |
| Nb | ppm | 9 | 7 | 18 | 8 | 10 | 8 | 12 | 17 | 17 | 14 | 21 | 15 | 20 |
| Mo | ppm | 6 | <2 | <2 | <2 | <2 | <2 | <2 | <2 | <2 | <2 | 2 | 5 | 2 |
| Ag | ppm | <0.5 | <0.5 | <0.5 | <0.5 | <0.5 | <0.5 | <0.5 | <0.5 | <0.5 | <0.5 | <0.5 | <0.5 | <0.5 |
| In | ppm | 0.2 | <0.2 | <0.2 | <0.2 | <0.2 | <0.2 | <0.2 | <0.2 | <0.2 | <0.2 | <0.2 | <0.2 | <0.2 |
| Sn | ppm | 1 | 4 | 3 | 1 | 1 | 1 | 4 | 5 | 6 | 3 | 14 | 7 | 5 |
| Sb | ppm | <0.5 | <0.5 | <0.5 | <0.5 | <0.5 | <0.5 | <0.5 | <0.5 | <0.5 | <0.5 | <0.5 | <0.5 | <0.5 |

**Table 3.** *Cont.*

| Sample No. | | Van Canh Plutonic Rock | | | | | | Hai Van Granitic Rock | | | | | | |
|---|---|---|---|---|---|---|---|---|---|---|---|---|---|---|
| | | **VN401** | **VN402** | **VN403** | **VN404** | **VN405** | **VN406** | **VN407** | **VN408** | **VN409** | **VN410** | **VN411** | **VN412** | **VN413** |
| Cs | ppm | 1.1 | 5.2 | 4.6 | 1.3 | 2.6 | 2.4 | 8.1 | 9.7 | 9.8 | 8.8 | 24.8 | 16.9 | 19.6 |
| La | ppm | 61.8 | 24.2 | 25 | 13.9 | 22.7 | 34.5 | 29.1 | 37.2 | 39.5 | 44.3 | 46.3 | 44.4 | 32 |
| Ce | ppm | 134 | 40 | 45 | 26.5 | 49.7 | 51.9 | 57.3 | 71.8 | 75.7 | 89.6 | 102 | 85.8 | 62.2 |
| Pr | ppm | 16.4 | 3.94 | 5.44 | 3.24 | 6.86 | 5.36 | 6.77 | 8.41 | 8.73 | 10.4 | 12 | 9.89 | 6.93 |
| Nd | ppm | 63.6 | 12.9 | 21.5 | 12.7 | 27.2 | 19 | 25.4 | 31 | 31.2 | 38 | 42.3 | 36.2 | 26 |
| Sm | ppm | 15.4 | 2.3 | 5.2 | 2.3 | 5 | 3 | 5.6 | 6.8 | 6.3 | 8.2 | 10.9 | 7.3 | 5.3 |
| Eu | ppm | 0.69 | 0.42 | 1.4 | 0.8 | 1.42 | 0.95 | 1.36 | 1.09 | 1.14 | 1.26 | 0.44 | 1.11 | 0.91 |
| Gd | ppm | 12.8 | 2.1 | 5 | 2 | 4.1 | 2.3 | 5 | 6 | 5.3 | 7.4 | 10 | 6.1 | 5.4 |
| Tb | ppm | 2 | 0.3 | 0.8 | 0.3 | 0.6 | 0.3 | 1 | 0.9 | 0.7 | 1.2 | 1.6 | 0.9 | 1 |
| Dy | ppm | 10.1 | 2.1 | 4.8 | 1.5 | 3.4 | 1.6 | 6.4 | 5.6 | 4 | 7.2 | 9.3 | 4.9 | 6.9 |
| Ho | ppm | 1.7 | 0.5 | 0.9 | 0.3 | 0.6 | 0.3 | 1.4 | 1.1 | 0.7 | 1.5 | 1.6 | 0.9 | 1.5 |
| Er | ppm | 4.1 | 1.5 | 2.4 | 0.8 | 1.9 | 0.9 | 4.5 | 3 | 1.8 | 4.2 | 4.1 | 2.5 | 4.6 |
| Tm | ppm | 0.51 | 0.26 | 0.34 | 0.12 | 0.26 | 0.13 | 0.73 | 0.41 | 0.25 | 0.61 | 0.54 | 0.34 | 0.67 |
| Yb | ppm | 2.9 | 1.8 | 2.2 | 0.8 | 1.7 | 0.8 | 5 | 2.7 | 1.7 | 3.8 | 3.3 | 2.2 | 4.6 |
| Lu | ppm | 0.4 | 0.31 | 0.38 | 0.14 | 0.25 | 0.16 | 0.76 | 0.44 | 0.27 | 0.62 | 0.46 | 0.35 | 0.73 |
| Hf | ppm | 5.4 | 2.9 | 7.8 | 4.8 | 4.3 | 4.5 | 4.2 | 5.3 | 5.7 | 6.9 | 5.8 | 5.9 | 4.8 |
| Ta | ppm | 1.1 | 1.8 | 1.3 | 0.9 | 0.8 | 0.8 | 1.6 | 2.2 | 1.8 | 2 | 3.7 | 2.5 | 2 |
| Tl | ppm | 0.5 | 0.8 | 0.8 | 0.1 | 0.2 | 0.2 | 0.6 | 0.9 | 1.1 | 1 | 2.4 | 1.2 | 1.3 |
| Pb | ppm | 21 | 22 | 31 | <5 | <5 | 8 | 26 | 22 | 31 | 23 | 43 | 35 | 15 |
| Bi | ppm | <0.4 | <0.4 | <0.4 | <0.4 | <0.4 | <0.4 | <0.4 | <0.4 | <0.4 | <0.4 | 0.4 | 0.9 | <0.4 |
| Th | ppm | 24.6 | 16.3 | 10.7 | 3 | 4 | 10 | 11.1 | 16.1 | 18.1 | 24.9 | 46.3 | 22.7 | 13.7 |
| U | ppm | 3.4 | 4.8 | 3.9 | 1.1 | 1.3 | 3.2 | 3.6 | 6.5 | 4.4 | 3.1 | 25.3 | 6.7 | 8.3 |

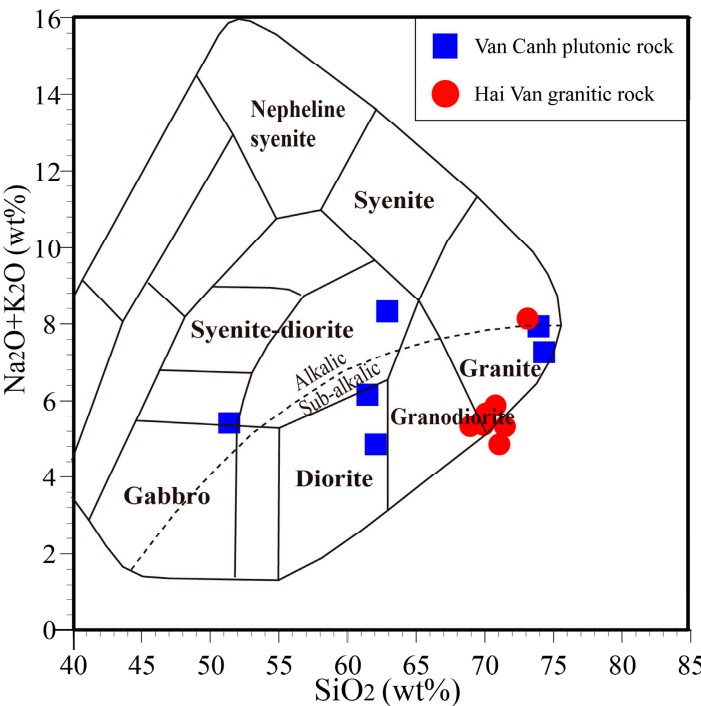

**Figure 5.** Na$_2$O + K$_2$O vs. SiO$_2$ (TAS) diagram for a classification of the collected plutonic rock samples [22,23]. The dashed line in the figure is the boundary between alkaline rocks and sub-alkaline rocks.

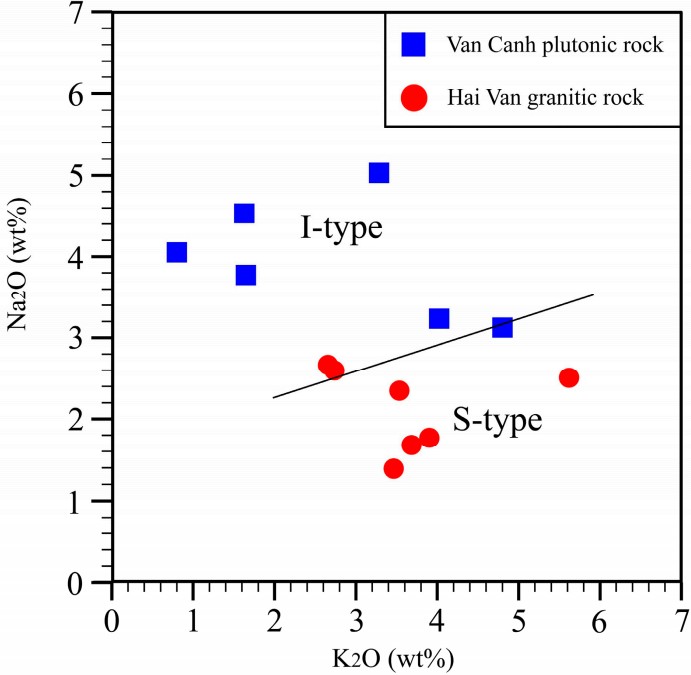

**Figure 6.** Na$_2$O vs. K$_2$O diagram for the classification of I- and S-type [24].

Based on the Zr vs. 10,000·Ga/Al diagram (Figure 8), many plutonic rocks are plotted near the boundary between I&S- and A-type. This result is consistent with the presence of plutonic rocks enriched in total alkali [26].

In contrast, on the basis of the Rb vs. Yb + Ta diagram (the tectonic setting classification diagram) (Figure 9), the Van Canh plutonic rocks in Quang Nam Province are all classified as volcanic arc granitic rocks [27]. In contrast, many of the Hai Van granitic rocks in Da Nang Province are categorized as syn-collision granitic rocks, but there are also some rocks within the volcanic arc granitic rock region.

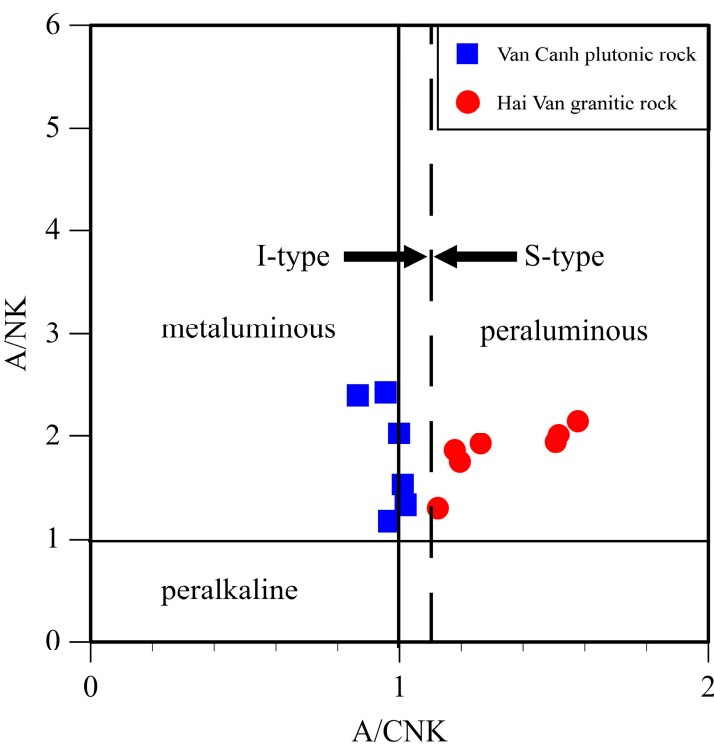

**Figure 7.** $Al_2O_3/(Na_2O + K_2O)$ vs. $Al_2O_3/(CaO + Na_2O + K_2O)$ diagram to classify peraluminous, mataluminous, and peralkaline rocks [24,25]. The dashed line shows the boundary between I- and S-type plutonic rocks.

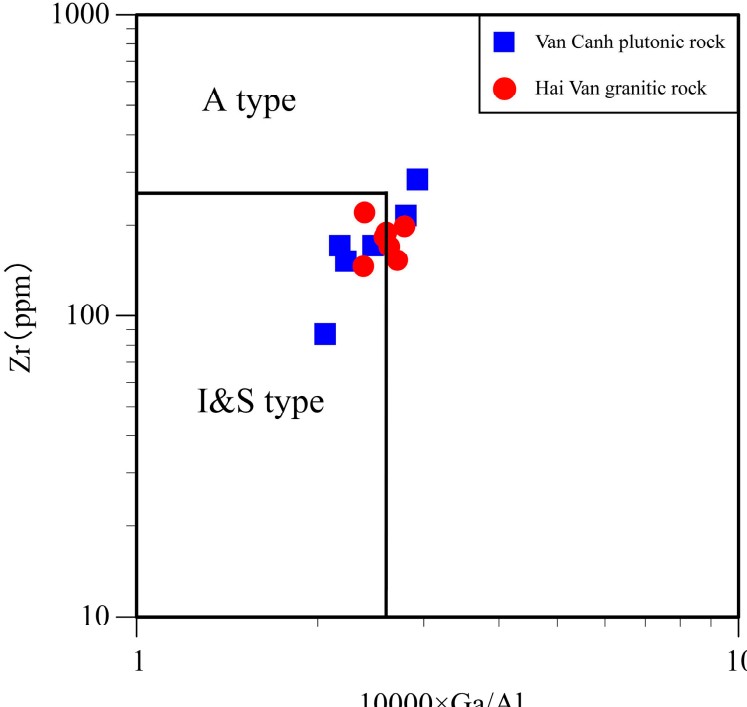

**Figure 8.** Zr vs. 10,000·Ga/A diagram to classify I&S- and A-type rocks [26].

On the basis of the Sr/Y vs. Y diagram to distinguish adakitic rocks from non-adakitic (calc-alkaline) rocks (Figure 10) [28], all the granitic rocks in Da Nang Province fall into the non-adakitic rock category. In contrast, the three Van Canh plutonic rocks in Quang Nam Province are adakitic rocks, with the remaining three classified as non-adakitic rocks.

However, these three non-adakitic rocks in Quang Nam Province also exhibit slightly high Sr/Y ratios compared to the Hai Van granitic rocks in Da Nang Province.

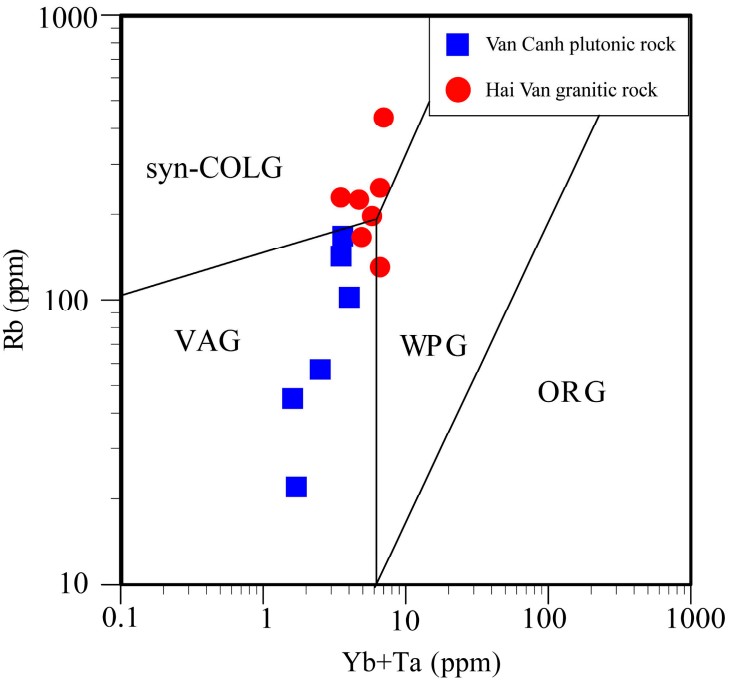

**Figure 9.** Rb vs. Yb + Ta tectonic setting classification diagram for the collected plutonic rock samples [27]. Abbreviations: syn-COL, syn-collision granitic rocks; WPG, within plate granitic rocks; VAG, volcanic arc granitic rocks; ORG, ocean ridge granitic rocks.

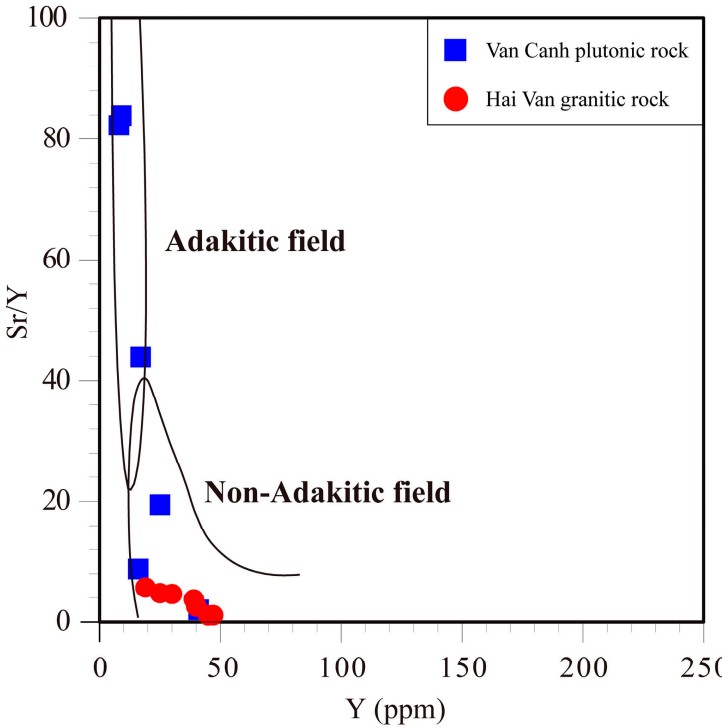

**Figure 10.** Sr/Y vs. Y diagram to classify adakitic rocks from non-adakitic rocks [28].

### 4.4. Chondrite-Normalized Rare Earth Element Pattern

Figure 11 shows the rare earth element (REE) pattern diagram, which was normalized using chondrite [29]. All the Hai Van granitic rocks in Da Nang Province exhibited a

negative Eu anomaly, whereas none of the Van Canh plutonic rocks in Quang Nam Province, except for one sample (VN401), had a significantly negative Eu anomaly.

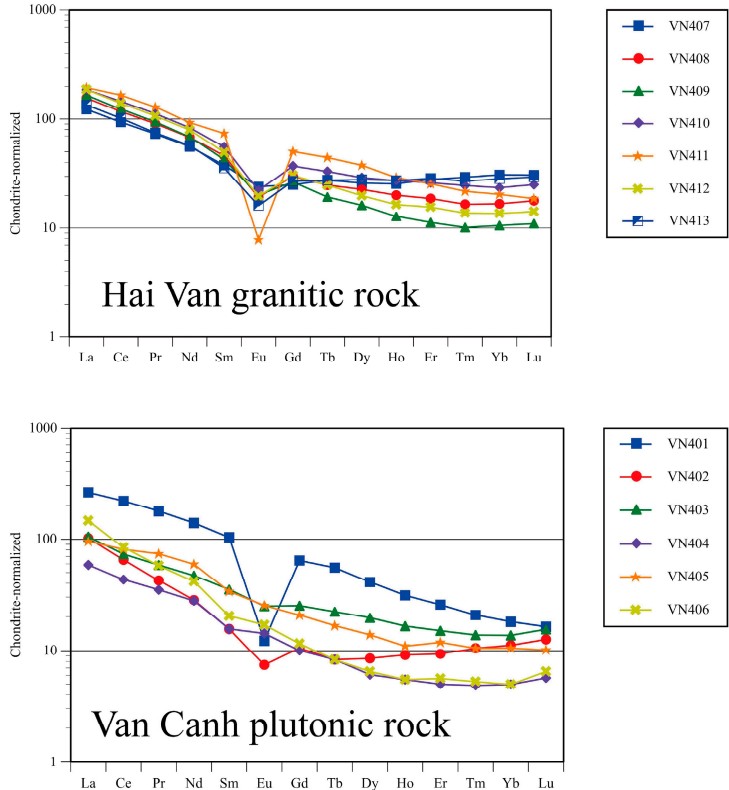

**Figure 11.** Chondrite-normalized rare earth element pattern [29]. The chemical composition data for the chondrite is from McDonough and Sun [30].

*4.5. Chemical Composition Analysis of Biotite*

The results of the analysis are shown in the Mg/(Mg + Fe) molar ratio vs. total Al diagram (Figure 12) and the $Al_2O_3$ vs. MgO diagram (Figure 13) [31]. It is evident that there are distinct compositional differences between the Van Canh plutonic rocks in Quang Nam Province (I-type) and the Hai Van granitic rocks in Da Nang Province (S-type). The biotite in the former exhibits lower total Al contents and higher Mg/(Mg + Fe) molar ratios when compared with the biotite in the latter.

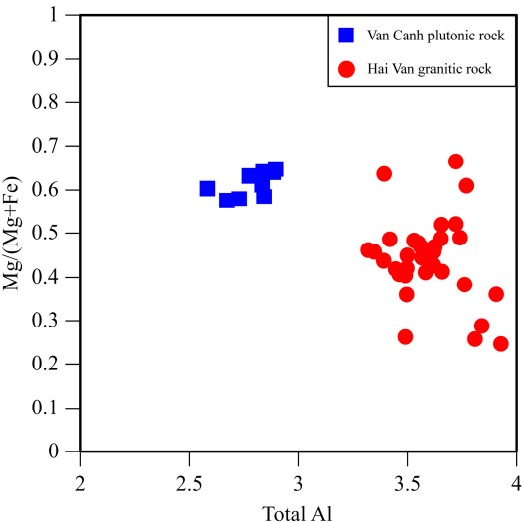

**Figure 12.** Mg/(Mg + Fe) molar ratio vs. total Al diagram for biotite based on O = 22.

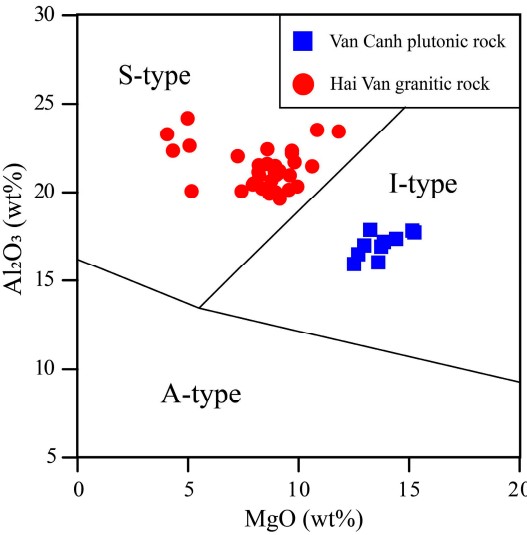

**Figure 13.** $Al_2O_3$ vs. MgO diagram for biotite [31].

*4.6. Nd and Sr Isotope*

The isotope ratio measurement results for Nd and Sr are shown in Table 4.

The $^{143}Nd/^{144}Nd$ and $^{87}Sr/^{86}Sr$ ratios for the Hai Van granitic rocks in Da Nang Province were 0.500755–0.512122 and 0.750000–0.820278, respectively (Table 4). In contrast, for the Van Canh plutonic rocks in Quang Nam Province, the $^{143}Nd/^{144}Nd$ and $^{87}Sr/^{86}Sr$ ratios were 0.512355–0.512422 and 0.706860–0.721638, respectively (Table 4).

Using an intermediate U-Pb age of 233 Ma for zircons from the Hai Van granitic rocks (242–224 Ma) and an intermediate U-Pb age of 240 Ma for those from the Van Canh plutonic rocks (251–229 Ma) [1,6], the Nd and Sr initial isotope ratios were determined to be 0.511561–0.511874 and 0.728026–0.792164 for the Hai Van granitic rocks in Da Nang Province, respectively, and 0.512115–0.512240 and 0.701999–0.711747 for the Van Canh plutonic rocks in Quang Nam Province, respectively (Table 4 and Figure 14).

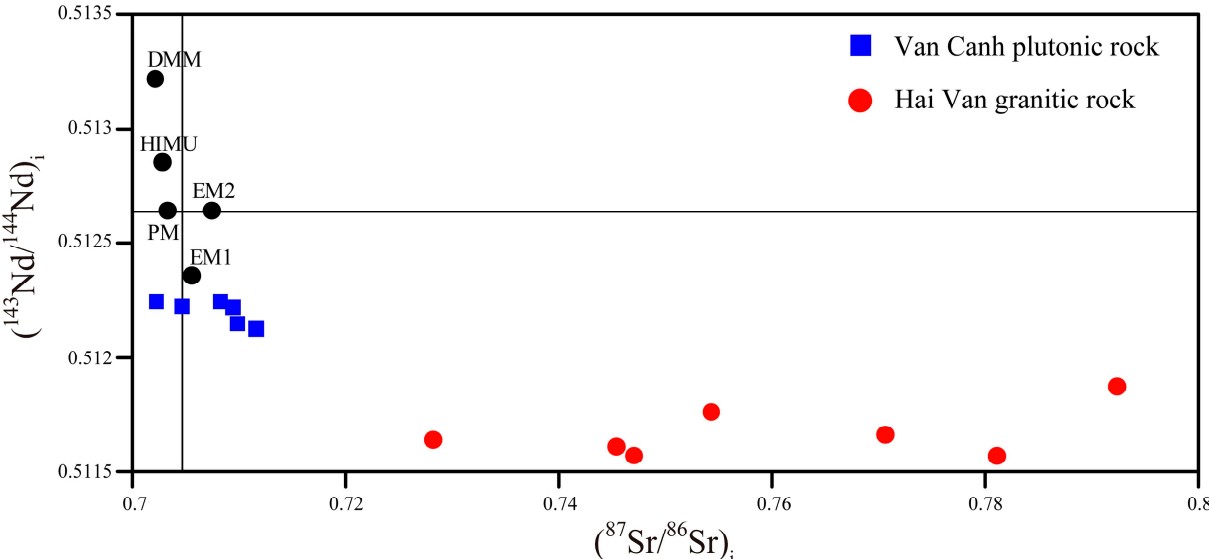

**Figure 14.** Diagram showing Nd vs. Sr initial isotope ratios for the collected plutonic rocks; calculated using the U-Pb ages of 240 Ma for zircons from the Van Canh plutonic rocks [6] and 233 Ma for those from the Hai Van granitic rocks [1]. The Nd and Sr initial isotope ratios (black dot) for the primitive mantle (PM), depleted MORB mantle (DMM), high μ mantle (HIMU), enriched mantle 1 (EM1), and enriched mantle 2 (EM2) [18,32] are plotted in the figure.

**Table 4.** Nd and Sr initial isotope ratios for the collected plutonic rocks.

| | Sample No. | U-Pb Age (Ma) | $^{87}Sr/^{86}Sr$ | $\pm 1\sigma$ | $^{87}Rb/^{86}Sr$ | $(^{87}Sr/^{86}Sr)_i$ | $^{143}Nd/^{144}Nd$ | $\pm 1\sigma$ | $^{147}Sm/^{144}Nd$ | $(^{143}Nd/^{144}Nd)_i$ |
|---|---|---|---|---|---|---|---|---|---|---|
| Van Canh plutonic rock | VN401 | | 0.721638 | 0.000011 | 3.47310 | 0.709782 | 0.512381 | 0.000009 | 0.152702 | 0.512141 |
| | VN402 | | 0.719586 | 0.000008 | 3.34661 | 0.708161 | 0.512411 | 0.000012 | 0.112440 | 0.512234 |
| | VN403 | 240 | 0.713560 | 0.000010 | 0.5311 | 0.711747 | 0.512355 | 0.000007 | 0.152526 | 0.512115 |
| | VN404 | | 0.706860 | 0.000009 | 0.65352 | 0.704629 | 0.512396 | 0.000005 | 0.114211 | 0.512217 |
| | VN405 | | 0.707037 | 0.000011 | 1.47577 | 0.701999 | 0.512422 | 0.000006 | 0.115928 | 0.512240 |
| | VN406 | | 0.713644 | 0.000012 | 1.22189 | 0.709473 | 0.512374 | 0.000010 | 0.099575 | 0.512218 |
| Hai Van granitic rock | VN407 | | 0.751066 | 0.000009 | 6.95216 | 0.728026 | 0.511855 | 0.000008 | 0.139021 | 0.511643 |
| | VN408 | | 0.750000 | 0.000010 | 0.968008 | 0.746792 | 0.511774 | 0.000005 | 0.138314 | 0.511563 |
| | VN409 | | 0.784132 | 0.000025 | 0.989624 | 0.780852 | 0.511755 | 0.000006 | 0.127322 | 0.511561 |
| | VN410 | 233 | 0.756568 | 0.000011 | 0.74888 | 0.754086 | 0.511963 | 0.000008 | 0.136072 | 0.511756 |
| | VN411 | | 0.820278 | 0.000018 | 8.48318 | 0.792164 | 0.512122 | 0.000006 | 0.162495 | 0.511874 |
| | VN412 | | 0.762957 | 0.000010 | 5.32060 | 0.745324 | 0.511799 | 0.000006 | 0.127156 | 0.511605 |
| | VN413 | | 0.817296 | 0.000036 | 14.14978 | 0.770403 | 0.511855 | 0.000005 | 0.128537 | 0.511659 |

## 5. Discussion

### 5.1. Geochemical Signatures of the Hai Van Granitic Rocks and the Van Canh Plutonic Rocks

The Triassic granitic rocks in Da Nang Province are the Hai Van granitic rocks, as reported by Hieu et al. [1], and are mainly classified as granite. They are classified as typical S-type granitic rocks based on the $Na_2O$ vs. $K_2O$ diagram, A/CNK molar ratio, biotite composition, and the presence of muscovite. In terms of magnetic susceptibility, they are classified as ilmenite series, which is consistent with their classification as S-type. In the tectonic setting classification diagram, most are classified as syn-collision granitic rocks. Furthermore, all of them are located near the boundary between I&S- and A-type. The chondrite-normalized REE patterns reveal negative Eu anomalies in all samples, indicating that crystallization differentiation of plagioclase occurred under reducing conditions. In terms of the chemical composition of biotite, the high total Al and a relatively low Mg/(Mg + Fe) molar ratio classify them as S-type (Figures 12 and 13), indicating that they originated from aluminum-rich sedimentary rocks. Furthermore, they exhibit typical characteristics of non-adakitic rocks (calc-alkaline rocks). The Sr initial isotope ratios of the Hai Van granitic rocks in Da Nang Province were determined to be very high (0.728026–0.792164). Considering these facts and the zircon U-Pb dating results from Hieu et al. [1] (242–224 Ma), the Hai Van granitic rocks in Da Nang Province are believed to have been generated by the collision between the South China and Indochina Blocks and originated from continental crustal materials (sedimentary rocks) (Figure 15).

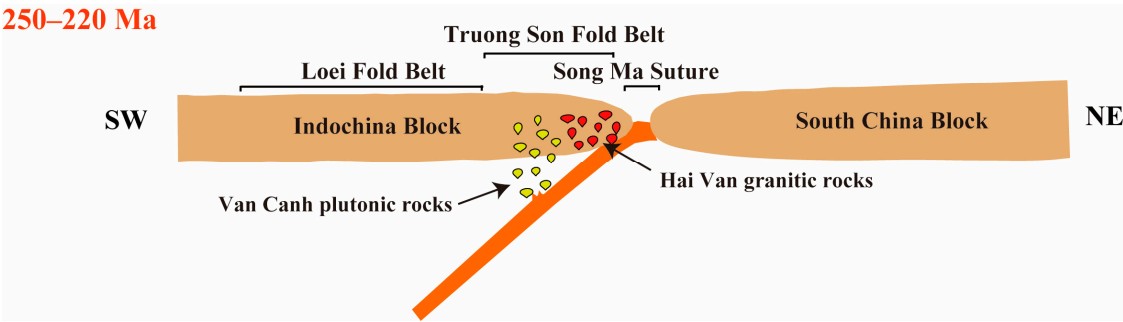

**Figure 15.** Schematic diagram showing the tectonic settings of the Van Canh plutonic rocks and the Hai Van granitic rocks in relation to the South China and Indochina Blocks.

In contrast, the Triassic Van Canh plutonic rocks in Quang Nam Province investigated in this study display a wide compositional range in the TAS diagram, including granite, diorite, and syenite-diorite. They are classified as typical I-type on the basis of the $Na_2O$ vs. $K_2O$ diagram, A/CNK molar ratio, biotite composition, and the presence of hornblende. They are classified as magnetite series, which is consistent with their classification as I-type. In the tectonic setting classification diagram, all of them are classified as volcanic arc granitic rocks. The REE patterns show slightly to no negative Eu anomaly, except for sample VN401. Regarding the chemical composition of biotite, the low total Al and high Mg/(Mg + Fe) molar ratios indicate that they are I-type. Among the six Van Canh plutonic rock samples, three samples are adakitic rocks with I-type features, and the other three are located within the non-adakitic rock range, with all but one showing a relatively high Sr/Y ratio. This suggests that, fundamentally, these plutonic rocks were formed through the involvement of adakitic magmas produced by the subduction of the high-temperature Song Ma Ocean beneath the Indochina Block. In addition, the Sr initial isotope ratios were determined to be low (0.701999–0.711747) and reveal the involvement of mantle materials in the genesis. Hieu et al. [1] described the plutonic rocks in Quang Nam Province as the Hai Van granitic rocks, which is a complete misidentification. Based on the magnetic susceptibility and geochemical signatures, they should be classified as the Van Canh plutonic rocks [16].

*5.2. Geological Settings*

Both the Hai Van granitic rocks and the Van Canh plutonic rocks studied here are situated in the Truong Son Fold Belt, which was generated in relation to the subduction of the Song Ma Ocean, part of the Paleo-Tethys Ocean that existed between the South China and Indochina Blocks, beneath the Indochina Block (Figure 15). The collision remnants of the South China and Indochina Blocks are preserved as the Song Ma Suture Zone. The Hai Van granitic rocks were formed in close proximity to the Song Ma Suture Zone, whereas the Van Canh plutonic rocks were formed in a more distant location. Based on the geochemical characteristics of both plutonic rocks, the Hai Van granitic rocks are considered to have been generated within a compressional field associated with the collision between the South China and Indochina Blocks, whereas the Van Canh plutonic rocks were formed in a back-arc-like environment associated with the subduction of the Song Ma Ocean, which is characterized by an extensional field where magma generated in the mantle rises more easily. It is also inferred that the ascent rate was faster, and there was a relatively low incorporation of continental crustal materials. The Van Canh and Ben Giang-Que Son plutonic rocks distributed in Kontum and Gia Lai provinces, central Vietnam, were considered to have been formed in the same tectonic settings [16]. These rocks are located farther from the Song Ma Suture Zone than the plutonic rocks investigated in this study. A similar scenario is observed between the Indochina and Sibumasu blocks [4,33]. The presence of the Sukhothai-Kampong Chhnang Zone, which contains ilmenite series and I-type granitic rocks with a significant influence of continental crustal materials, is attributed to the subduction of the Paleo-Tethys Ocean beneath the Indochina Block. Moving eastward, away from the Paleo-Tethys Ocean, the formation of a back-arc basin (Loei Fold Belt) is evident, where magnetite series and I-type granitic rocks were formed; these rocks exhibit adakitic characteristics. The phenomena between the South China and Indochina Blocks are more pronounced between the Sibumasu and Indochina Blocks. This difference is considered to be related to the size of the Paleo-Tethys Ocean between these blocks; the Paleo-Tethys Ocean between the Sibumasu and Indochina Blocks was larger, suggesting that the subducted oceanic crust was also larger in this region.

## 6. Conclusions

The following conclusions were obtained for the plutonic rocks investigated in this study:

(1) The Hai Van granitic rocks in Da Nang Province exhibit a typical granite composition and are clearly classified as S-type and ilmenite series. They indicate crystallization differentiation of plagioclase under reducing conditions. The high Sr initial isotope ratios suggest the involvement of the continental crustal materials. Based on previously reported U-Pb age results, it is believed that the granitic rocks were generated during the collision of the South China and Indochina Blocks.

(2) The plutonic rock samples in Quang Nam Province exhibit a wide range of $SiO_2$ contents and are classified as magnetite series and I-type. They show little evidence of a negative Eu anomaly. Many samples exhibited characteristics of adakitic rocks. The low Sr initial isotope ratios suggest the involvement of mantle materials. The adakitic magma was likely formed by partial melting of the relatively hot Song Ma Ocean as it was subducted beneath the Indochina Block. Thus, the plutonic rocks in this region are considered to be the Van Canh plutonic rocks based on their geochemical similarities.

(3) The Hai Van granitic rocks in Da Nang Province and the Van Canh plutonic rocks in Quang Nam Province are situated in the Truong Son Fold Belt. It is speculated that the former was formed in a compressional field through the subduction of the Song Ma Ocean, while the latter is believed to have been formed in an extensional field corresponding to a back-arc-like environment associated with the subduction of the Song Ma Ocean.

**Author Contributions:** Conceptualization, E.U.; methodology, E.U.; formal analysis, E.U. and T.Y.; investigation, E.U. and T.Y.; resources, E.U. and T.Y.; data curation, E.U. and T.Y.; writing—original draft preparation, E.U.; writing—reviewing and editing, E.U. and T.Y.; visualization, E.U. and T.Y.;

supervision, E.U.; project administration, E.U.; funding acquisition, E.U. All authors have read and agreed to the published version of the manuscript.

**Funding:** This research was funded by a Waseda University Grant for Special Research Projects: 2022R-015.

**Data Availability Statement:** All data are included/referenced in this article.

**Acknowledgments:** This research was in part supported by a Joint Research Grant for Environmental Isotope Study of the Research Institute for Humanity and Nature. We would like to express our gratitude for the valuable comments received from three anonymous reviewers on the manuscript of this paper.

**Conflicts of Interest:** The authors declare no conflict of interest.

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
