# Peer review of "Differences in Geochemical Characteristics and Tectonic Settings between Hai Van Granitic Rocks in Da Nang Province and Van Canh Plutonic Rocks in Quang Nam Province, Central Vietnam"

_geosciences, doi:10.3390/geosciences14010013_

Round 1

Reviewer 1 Report

Comments and Suggestions for Authors

This paper presents interesting magnetic and geochemical dataset of granitic rocks respectively in Da Nang and Quang Nam Provinces, Central Vietnam, with aims to discuss its petrogenesis and tectonic settings. The topic is suitable for Geosciences. There also exist some flaws through the manuscript, which should be clarified and solved before it can be considered for publication.

1 The Van Canh granitic rocks contain samples with SiO2 even as low as 51.38%, which cannot be regarded as granitic rock. In this respect, the title should also be revised to correspond to the corrected lithological assemblage.

2 The measured Sr isotopic compositions in the I-type granitic rocks are problematic and meaningless, e.g., some samples contain initial 87Sr/86Sr ratio even lower than 0.702, a value of modern mid-oceanic ridge basalts (MORB). The Sr-Nd isotope diagram (Fig. 13) has no geological significance yet.

3 You should recalculate the Nd isotopic ratios as initial epsilon Nd value instead of the currently initial 143Nd/144Nd. The comparison of initial epsilon Nd value is more prevalent in modern geochemistry.

4 It remains unclear why the authors consider the formation of I-type granitic rocks in a back-arc basin. You should provide more geological and geochemical constraints to support such a hypothesis.

5 The discussion about the petrogenesis of both rock types is surficial. More details should be presented in the related parts.

Author Response

To Reviewer 1

Thank you for your insightful comments and suggestions to our manuscript submitted to Geosciences. We revised our manuscript taking them into consideration as follows:

Point 1: The Van Canh granitic rocks contain samples with SiO2 even as low as 51.38%, which cannot be regarded as granitic rock. In this respect, the title should also be revised to correspond to the corrected lithological assemblage.

Response: Considering the content of SiO2, we have revised to use the following terms:” the Van Canh plutonic rocks” and “the Hai Van granitic rocks”.

Point 2: The measured Sr isotopic compositions in the I-type granitic rocks are problematic and meaningless, e.g., some samples contain initial 87Sr/86Sr ratio even lower than 0.702, a value of modern mid-oceanic ridge basalts (MORB). The Sr-Nd isotope diagram (Fig. 13) has no geological significance yet.

Response: The influence of continental crustal material is significant in the formation of the Hai Van granitic rocks, as indicated by the high values of 87Sr/86Sr initial ratio. Conversely, it is evident that the generation of the Van Canh plutonic rocks is strongly influenced by mantle materials. In other words, the distinct origins of these two rock types are highlighted, emphasizing the utility of Sr isotope ratios as valuable indicators.

Point 3: You should recalculate the Nd isotopic ratios as initial epsilon Nd value instead of the currently initial 143Nd/144Nd. The comparison of initial epsilon Nd value is more prevalent in modern geochemistry.

Response: Regarding Nd isotope ratios, some researchers use the initial epsilon Nd value, while others use the initial 143Nd/144Nd value. In our previous papers, we have used the latter, and for the sake of consistency, we will continue to use the initial 143Nd/144Nd value in this paper as well.

Point 4: It remains unclear why the authors consider the formation of I-type granitic rocks in a back-arc basin. You should provide more geological and geochemical constraints to support such a hypothesis.

Response: We replaced "a back-arc basin" with "a back-arc-like environment."

Point 5: The discussion about the petrogenesis of both rock types is surficial. More details should be presented in the related parts.

Response: In the discussion, it is clearly stated that the Hai Van granitic rocks distributed in Da Nang Province, due to their geochemical characteristics, are S-type granites generated by the collision between the South China and Indochina blocks. In contrast, plutonic rocks distributed in Quang Nam Province, previously considered as the Hai Van granitic rocks by Hieu et al. (2015), have been revealed to be Van Canh plutonic rocks. These Van Canh plutonic rocks exhibit geochemical features of adakitic rocks and are formed through subduction of the Song Ma Ocean beneath the Indochina Block. The discussion emphasizes that the Hai Van granitic rocks and the Van Canh plutonic rocks are distinctly different in geochemical signatures and also in their tectonic settings. We consider that the discussion is considered comprehensive, without lacking in content, and is by no means superficial.

Reviewer 2 Report

Comments and Suggestions for Authors

Author Response

To Reviewer 2

Thank you for your insightful comments and suggestions to our manuscript submitted to Geosciences. We revised our manuscript taking them into consideration as follows:

Point 1: In chemical analyses: where is the LOI?

Response: LOIs were added to Table 3.

Point 2: Also, I noticed some samples have a total more than 100% without the LOI. Please, check the quality of the chemical analyses.

Response: Even including the LOIs, the total is still below 100.6%. Therefore, we do not believe that there is a problem with the reliability of the analytical values.

Point 3: The authors commented on figure 5 and mention that "The granitic rocks from both

regions include alkalic rocks". It is not true most of the analyzed samples are sub-alaklic .

Response: Indeed, most rocks are sub-alkalic, but three samples are alkalic, so we have revised it accordingly.

Point 4: The section of Whole-rock chemical composition is very poor section and need to discuss the data in more detail.

Response: In the section on the whole-rock chemical composition, we believe that we have adequately described the necessary figures using the obtained whole-rock chemical composition to distinguish the geochemical signatures of the two plutonic rock suites under investigation.

Point 5: In the section of mineral composition: why did the authors analyzes only the biotite?

Response: The composition of biotite reflects whether it originates from continental crustal materials or mantle materials. Therefore, in this paper, we conducted an analysis focusing on biotite.

  An Al2O3 vs. MgO diagram has been added as Figure 13 for biotite to distinguish I-, S-, and A-type.

Point 6: The biotite in the Quang Nam Province analyzed from which rock type (granite, syenite or gabbro)?

Response: We conducted biotite analysis on the plutonic rocks in Quang Nam Province, specifically targeting granite, granodiorite, and diorite (refer to Table 2).

Point 7: Why did not the author represent the chemical analyses in the manuscript, although this manuscript is short paper?

Response: In the Results section, we have conducted various analyses using the results of chemical composition analysis. Based on the definition by MDPI, we are sure that based on word count, this paper falls under the category of an “article” rather than a “short paper”.

Point 8:  Why did not the authors use the biotite chemical analyses to deduce the magma type and tectonic setting of the studied granitic rocks.

Please, use the following references to deduce magma type and tectonic setting for the studied rocks:

(1)- Abdel-Rahman, A.M., 1994. Nature of biotites from alkaline, calc-alkaline and

peraluminous magmas. Journal of Petrology 35, 525–541.

(2) Nachit, H., 1985. Composition chemique des biotites et typologie magmatique des

granitoids. Comtes Rendus Hebdomadaires de l'Academie des Sciences 301(11), 813-

818

Response: In the Discussion chapter, we discussed the differences in the origin of both rock suites (I-type or S-type) using the chemical composition of biotite.

Using the Al2O3 vs. MgO diagram for biotite based on Abdel-Rahman (1994), it was possible to clearly delineate the differences in I-, S-, and A-type characteristics for the Hai Van granitic rocks and the Van Canh plutonic rocks.

Reviewer 3 Report

Comments and Suggestions for Authors

I have carefully read the manuscript entitled “Differences in Geochemical Characteristics and Tectonic Settings between Hai Van Granitic Rocks in Da Nang Province and Van Canh Granitic Rocks in Quang Nam Provinces, Central Vietnam”. The datasets are eligible, but the authors show poor writing on the whole manuscript, especially in the introduction and discussion parts. Additionally, some of the samples are not granitic rocks, but the authors consider them equally as granitic rocks and did some discrimination and discussion, this is totally incorrect in principle. I list some general comments below, and unfortunately, I think this manuscript did not reach the standard for publication.

1. The Introduction is poor written. No scientific problem was proposed. The authors should tell the readers why they select this research object? What questions remain unclear or debated? And what did they figured out in this research.

2. In the sample description part, the estimated proportion of each rock-forming mineral should be listed. For example: quartz (30-40 %).

3. I suggest the authors unify the name of the granitic rocks, using Hai Van and Van Canh granitic rocks in the whole text, rather than using granitic rocks in Quang Nam Province or in Da Nang Province. The mixed expression makes readers chaos and confusing.

4. I noticed that some Van Canh rock samples have very low SiO2 contents, they are not belong to granitic rocks. So the subsequent discriminations for granitic rocks are disconfirm.  

5. In figure 6, simply using K2O and Na2O to discriminate I- and S-type granite is incorrect. There are a lot of exceptions, for example: highly evolved granites or A-type granites can also have high Na2O contents.

6. The discussion is also poor written without any subchapters. The whole discussion is extremely brief, and do not list any theories, only describe the results show on the figures.

7. The author did not consider the possibility of A-type granites or highly evolved granites. In fact, some of the Van Canh samples plot in the A-type field. To distinguish them from A-type and I-type granites, the crystallization temperature of granites should be estimated.

8. I-type granites and adakites are two different things. Better describe them as adakites with I-type features or I-type granites with adakitic features.

9. The Hai Van granitic rocks have extremely high Sr isotopic composition, they also show moderate Eu depletion, high Rb contents, and muscovite exhibit as the rock-forming mineral. This is the features of either highly evolved granite or granite with strong melt-fluid interaction, which can also have high A/CNK values. The authors should also consider it carefully to distinguish them from S-type granite and highly evolved granite.

10. The conclusions are lengthiness, it seem more like a discussion rather than a conclusion.

Comments on the Quality of English Language

The authors should polish the manuscript by native speakers, some expressions in the text are difficult to understand.

Author Response

To Reviewer 3

Thank you for your insightful comments and suggestions to our manuscript submitted to Geosciences. We revised our manuscript taking them into consideration as follows:

Point 1. The Introduction is poor written. No scientific problem was proposed. The authors should tell the readers why they select this research object? What questions remain unclear or debated? And what did they figure out in this research.

Response: As mentioned in the Introduction chapter, we identified the presence of the Van Canh plutonic rocks in the region investigated by Hieu et al. (2015), in addition to the Hai Van granitic rocks. Therefore, the primary objective of this paper is to elucidate this fact based on the results of various chemical analysis and physical measurement. We believe that the purpose of this paper is clearly defined in the Introduction chapter.

Point 2. In the sample description part, the estimated proportion of each rock-forming mineral should be listed. For example: quartz (30-40 %).

Response: In the footnotes of Table 2, there are descriptions related to quantities such as circles (〇: 30-10 vol.%) and triangles (△: 10-2 vol.%), allowing for a general understanding of the quantities of each mineral.

Point 3. I suggest the authors unify the name of the granitic rocks, using Hai Van and Van Canh granitic rocks in the whole text, rather than using granitic rocks in Quang Nam Province or in Da Nang Province. The mixed expression makes readers chaos and confusing.

 I noticed that some Van Canh rock samples have very low SiO2 contents, they are not belonging to granitic rocks. So the subsequent discriminations for granitic rocks are disconfirm. 

Response: Considering the content of SiO2, we have revised to use the following terms: the Van Canh plutonic rocks (in Quang Nam Province) and the Hai Van granitic rocks (in Da Nang Province).

Point 4. In figure 6, simply using K2O and Na2O to discriminate I- and S-type granite is incorrect. There are a lot of exceptions, for example: highly evolved granites or A-type granites can also have high Na2O contents.

Response: To distinguish between I- and S-type, in addition to the Na2O vs. K2O diagram, we also employed the A/NK vs. A/CNK diagram. We believe that the Van Canh plutonic rocks and the Hai Van granitic rocks were clearly classified using these diagrams.

Point 5. The discussion is also poor written without any subchapters. The whole discussion is extremely brief, and do not list any theories, only describe the results show on the figures.

Response: We divided the Discussion chapter into two sections. In Section 5.1, we summarized the differences in geochemical signatures between the Hai Van granitic rocks and the Van Canh plutonic rocks, clearly delineating the distinctions in their origins. In Section 5.2, we discussed the tectonic settings of both rock suites, making the differences clear.

Point 6. The author did not consider the possibility of A-type granites or highly evolved granites. In fact, some of the Van Canh samples plot in the A-type field. To distinguish them from A-type and I-type granites, the crystallization temperature of granites should be estimated.

Response: Based on the TAS diagram and discriminant diagram for I&S- and A-type, it was determined that at least the Van Canh plutonic rocks, VN401 and VN403, are classified as A-type. Therefore, this information has been described in the manuscript.

Point 7. I-type granites and adakites are two different things. Better describe them as adakites with I-type features or I-type granites with adakitic features.

Response: In the manuscript, the expression "adakitic rocks with I-type features" was used

Point 8. The Hai Van granitic rocks have extremely high Sr isotopic composition, they also show moderate Eu depletion, high Rb contents, and muscovite exhibit as the rock-forming mineral. This is the features of either highly evolved granite or granite with strong melt-fluid interaction, which can also have high A/CNK values. The authors should also consider it carefully to distinguish them from S-type granite and highly evolved granite.

Response: Due to the low SiO2 content ranging from 69-73 wt% and a relatively low Na2O+K2O content of around 5%, the Hai Van granitic rocks are considered S-type granites, but they are not believed to be highly evolved granites.

Point 9. The conclusions are lengthiness; it seems more like a discussion rather than a conclusion.

Response: We made efforts to shorten the Conclusions chapter as much as possible.

Round 2

Reviewer 2 Report

Comments and Suggestions for Authors

No

Author Response

To Reviewer 2

Thank you for your kind revision of our manuscript submitted to Geosciences.

Reviewer 3 Report

Comments and Suggestions for Authors

The current version of the manuscript has been largely improved, I suggest it can be accepted after minor revision. However, There are still several points to be further improved.

1. The disscusion part is too simple, I suggest the authors add some chapters to discuss the formation age, source region and protolith in detail.

2. The conclusions are still to long, simplified them into several sentences.

3.  Van Canh rock samples which not belonging to granitic rocks are not suitable for current diagrams, delete them.

4. In the introduction part, the authors should clarify what scientific questions remain unsolved? Or why they choose to study these rocks?

Comments on the Quality of English Language

The English Language has been improved and only need minor polishment.

Author Response

To Reviewer 3

Thank you again for your insightful comments and suggestions to our manuscript. We revised the manuscript taking them into consideration as follows:

Point 1. The discussion part is too simple, I suggest the authors add some chapters to discuss the formation age, source region and protolith in detail.

Response: The ages of the Hai Van granitic rocks and the Van Canh plutonic rocks have already been determined using zircon U-Pb dating by Hieu et al. (2015) and Hung et al. (2022), respectively. Both studies have obtained Triassic ages, and we believe there is no need to treat them as independent sections.

  The source region and protolith have already been discussed in section 5.1 of the Discussion, chapter so we believe there is no need to add further information on this matter.

Point 2. The conclusions are still too long, simplified them into several sentences.

Response: We have made every effort to shorten the Conclusions as much as possible.

Point 3.  Van Canh rock samples which not belonging to granitic rocks are not suitable for current diagrams, delete them.

Response: The Van Canh plutonic rocks are characterized by a wide range of SiO2 compositions, attributed to variations in the proportion of continental crustal materials captured by the magma. Therefore, we believe it is acceptable to include all samples in the figures.

Point 4. In the introduction part, the authors should clarify what scientific questions remain unsolved? Or why they choose to study these rocks?

Response: We added the following sentence to the Introduction chapter: “These two rock bodies are located within the Truong Son Fold Belt. Another objective of this study is to elucidate the formation mechanism of the Truong Son Fold Belt based on the difference in the formation process of the two rock bodies.”